# CircAFF1 Is a Circular RNA with a Role in Alveolar Rhabdomyosarcoma Cell Migration

**DOI:** 10.3390/biomedicines11071893

**Published:** 2023-07-04

**Authors:** Alvaro Centrón-Broco, Francesca Rossi, Chiara Grelloni, Raffaele Garraffo, Dario Dattilo, Andrea Giuliani, Gaia Di Timoteo, Alessio Colantoni, Irene Bozzoni, Manuel Beltran Nebot

**Affiliations:** 1Department of Biology and Biotechnology Charles Darwin, Sapienza University of Rome, 00185 Rome, Italy; 2The Babraham Institute, Babraham Research Campus, Cambridge CB22 3AT, UK; 3Center for Life Nano- & Neuro-Science, Fondazione Istituto Italiano di Tecnologia (IIT), 00161 Rome, Italy

**Keywords:** circAFF1, alveolar rhabdomyosarcoma, migration

## Abstract

Circular RNAs (circRNAs), covalently closed RNAs that originate from back-splicing events, participate in the control of several processes, including those that occur in the development of pathological conditions such as cancer. Hereby, we describe circAFF1, a circular RNA overexpressed in alveolar rhabdomyosarcoma. Using RH4 and RH30 cell lines, a classical cell line models for alveolar rhabdomyosarcoma, we demonstrated that circAFF1 is a cytoplasmatic circRNA and its depletion impacts cell homeostasis favouring cell migration through the downregulation of genes involved in cell adhesion pathways. The presented data underline the importance of this circular RNA as a new partial suppressor of the alveolar rhabdomyosarcoma tumour progression and as a putative future therapeutic target.

## 1. Introduction

Circular RNAs (circRNAs) are a class of single-stranded RNA molecules produced from back-splicing reactions. During these events, a 5′ splice site of a precursor RNA is linked to an upstream 3′ splice site, resulting in the formation of a covalently closed continuous loop [1].

In recent years, circRNAs arose from rare splicing phenomena to a large class of non-coding RNA transcripts. Circular RNAs are evolutionarily conserved among several species and their expression is tightly modulated in physiological and pathological processes [2,3,4,5].

Moreover, circRNAs can control gene expression through a myriad of mechanisms: they can sequester miRNA or proteins, decreasing their availability [6,7,8,9], and they can act as docking sites for multiple proteins [10,11], as templates for cap-independent translation transcripts [12,13,14,15], as recruiters of RNA-stabilising proteins [16] and as regulators of linear counterpart transcription [17].

CircRNAs roles and the regulation of their expression have been widely studied in several cellular processes, such as cell growth and differentiation, metabolic reprogramming, and cancer formation and development [18,19,20]. In the past few years, it has also been demonstrated that adenosine N6-methylation (m^6^A) RNA chemical modification can modulate circRNA biogenesis and translation [21,22].

Rhabdomyosarcoma (RMS) is the most abundant paediatric soft-tissue sarcoma, accounting for about one-twentieth of infants’ tumours [23]. The onset of the malignancy transformation is thought to be the alteration of the myogenic differentiation programme of mesenchymal cells transforming to skeletal muscle cell lineage [24]. RMS tumour grading and prognosis are based on histological architecture. Among the main subtypes, embryonal rhabdomyosarcoma (ERMS) and alveolar rhabdomyosarcoma (ARMS) account for three-fifths and one-fifth of all rhabdomyosarcoma cases [25].

ERMS is mainly developed in infants during the first ten years of life, whereas ARMS preferentially develops in teenagers and is usually categorised with a more adverse prognosis [25]. Overall 3-year survival depends on subtype and grading, generally exceeding 70% for low-risk RMS patients, while barely reaching 25–30% in high-risk cases [26]. The embryonal subtype is largely modelled using the RD cells, derived from pelvic embryonal RMS biopsy in a 7-year-old girl [27], whilst RH4 and RH30 cells model alveolar RMS and originated from a pulmonary metastasis of a 7-year-old female and a bone marrow metastasis 16-year-old male infants, respectively [27].

Here, we describe circAFF1, a circRNA abundantly expressed in several RMS cell lines with a role in controlling the migration and invasiveness capabilities of those rhabdomyosarcoma cancer cell lines.

## 2. Materials and Methods

### 2.1. Cell Cultures, Transfections and Flow Cytometric Analysis of Cell Cycle

Primary human myoblasts, ERMS RD cells, ARMS RH4 and RH30 cells were cultured as described previously [28]. Cell lines were previously used and authenticated as described in [29]. All cell lines were tested for mycoplasma contamination.

Transfections were performed using 30 nM siRNAs (Dharmacon, Lafayette, CO, USA) and Dharmafect-1 Transfection Reagent (Dharmacon Lafayette, CO, USA) according to the manufacturer’s instructions. Transfection media was replaced after 24 h with and cells were harvested 48 h post-transfection unless otherwise noted. The siRNAs used in this manuscript are provided in Appendix A.

Flow cytometric analysis of the cell cycle was performed using a BD FACSCalibur Flow Cytometer (BD Biosciences, Franklin Lakes, NJ, USA) after staining with 1 mg/mL of propidium iodide (Sigma), as detailed in [28].

### 2.2. RNA Purification and RNase-R Digestion

RNA of myoblasts, RD cells, ARMS RH4 and RH30 with or without siRNA treatment was obtained using the Direct-zol RNA kit and digested with DNAse-I (Zymo Research, Irvine, CA, USA) following the manufacturer’s instructions.

For cDNA generation, PrimeScript RT Kit (Takara Bio, Kusatsu, Japan) was used. For qRT-PCR experiments, Life-Tech SYBR (Life Technologies, Carlsbad, CA, USA) was used. Unless otherwise specified, RNA levels were calculated as 2^ΔΔCt^ relative to GAPDH mRNA, and the control sample was set as 1. MyTaq-HS DNA Polymerase (Bioline, London, UK) was used for non-quantitative amplifications according to the manufacturer’s protocol. RNase R (Epicentre, Madison, WI, USA) treatment was performed as described previously [13].

A detailed list of the primers used can be found in Appendix A.

### 2.3. Patient Biopsies

Tumour sections from 9 primary RMS tumours, 4 ARMSs and 5 ERMSs, were obtained at diagnosis time from teens admitted to the Department of Oncology at Alder Hey Children’s NHS Foundation Trust, Liverpool, UK. Control RNA was obtained from skeletal muscle biopsies from 3 teens enduring surgery for non-oncological disorders. Legal written informed consent was obtained, and the experiment acquired ethical review and approval (Alder Hey Children’s NHS Foundation Trust Ethics Committee, approval number 09/H1002/88).

### 2.4. Subcellular Fractionation

Nucleus and cytoplasm fractions were extracted as described in [28]. Reverse transcription was performed using iso-volume of both fractions of RNA. For chart depiction, the nuclear RNA amount was determined as 2^−ΔCt^ and then transformed into a percent fraction.

### 2.5. Growth Curve Assay

For each biological replicate, (*n* = 3) 7.5 × 10^4^ SCR, circAFF1 and AFF1 mRNA-depleted RH4 cells were seeded, and the number of alive cells was counted after 24 h, 48 h and 72 h using trypan blue as a contrast dye.

### 2.6. Sucrose Gradient Fractionation

Sucrose gradient fractionation of cytoplasmatic RD cell lysate was performed as described previously [13].

### 2.7. RNA Sequencing and Bioinformatic Analyses

Total RNA extraction and DNase treatment were performed as described previously. Three equivalent biological experiments of RH4 cells in control conditions (si-SCR) and two biological replicates of RH4 cells upon circAFF1 knockdown (si-circAFF1) were extracted for RNA sequencing as described before [28].

Data from differential gene expression analysis can be accessed in Appendix A.

### 2.8. Migration and Wound Healing Experiments

SCR, circAFF1, circAFF1#2, AFF1 mRNA, S100A2, and TLR4 depleted RH4 cells and SCR, circAFF1, AFF1 mRNA depleted RH30 cells were cultured in complete medium; 48 h after transfection 1.5 × 10^5^ cells per well were seeded into ThinCert^TM^ cell culture inserts with 8 μm pore translucid membranes (Vetro Scientifica, GR665638, Rome, Italy). Chambers with cells contained medium without serum, whilst the lower well had complete medium with FBS to act as a chemoattractant. After 15 h, cells in the top chamber were removed by scratching, and migrated cells were fixed and stained using crystal violet 4% or with 1 × 10^−3^ g/L DAPI staining (Sigma-Aldrich, Burlington, MA, USA) supplemented with 0.5% Triton^TM^ X-100 (Sigma-Aldrich). Cells were washed and imaged using an inverted microscope Zeiss Axio Observer A1 Phase Contrast. Pictures were taken using a Zeiss Plan-Neofluar 10× lens (NA 0.3) and gathered with the Zeiss AxioVision software REL.4.9.1 (Carl Zeiss AG, Oberkochen, Germany). Cells were counted in a minimum of 8 fields and the relative number of migrated cells per field was calculated as a fold change with respect to the experimental control sample set as 1. For wound healing, cells were transfected as described previously, and at 24 h post-transfection, wounding was performed using a 0.9 mm diameter scratcher. Samples were imaged at the same coordinates at 0 h and 24 h post-scratch (24 h and 48 h post-transfection) using an inverted microscope Zeiss Axio Observer A1 Phase Contrast supported with a Zeiss AxioCam MRm camera. The area of the wound was obtained using ImageJ software version 1.52p (National Institutes of Health (NIH), Bethesda, MD, USA). ΔArea was calculated from at least 7 points for each replicate of transfection and plotted using Prism^TM^.

### 2.9. Western Blot

Total cell extract for RH4 cells was obtained from lysing cells 48 h post-transfection with Total Protein Lysis Buffer (100 mM Tris pH 7.5, EDTA 1 mM, SDS 2%, PIC 1× (Complete-EDTA free, Roche—Merck, Basel, Switzerland). Lysates were agitated for 20 min on ice and spun at max speed in a cold centrifuge for clearing. Lysates (25–30 μg) were run in a Novex BT-acrylamide pre-casted gel (Thermo Fisher, Waltham, MA, USA) and electro-transferred to a nitrocellulose membrane. Blot procedures used the following antibodies: YAP1 (63.7) sc-101199, NOS1 (N20) sc-648, RUNX2 sc390351, TLR4 sc293072 and β-Actin-Peroxidase A3854. After secondary antibody incubation, imaging was performed using the ChemiDoc MP Imager (BioRad, Hercules, CA, USA) and analysed with Image Lab 6.1 software (BioRad, Hercules, CA, USA). For quantification, the volumetry of each blot was referred to β-actin as the loading control.

## 3. Results

### 3.1. CircAFF1 Is Abundantly Expressed in RMS-Derived Cell Lines

AFF1 (AF4/FMR2 family member 1) is a gene encoding a protein involved in the formation of several complexes in the cells such as the Super Elongation Complex (SEC), the P-TEFb complex and the ELL complex, with roles in the regulation of transcription and the osteoblastic differentiation process [30]. Curiously, AFF1 can also be strongly linked to cancer, as the chromosomal translocation of this *locus* generates fusion proteins associated with acute lymphoblastic leukaemia [31]. The AFF1 locus also can generate a circular RNA matching the annotated circular hsa_circ_0001423 (CircAFF1) [32,33]. This circular RNA is generated by the back-splicing of the third and fourth exons of the linear isoform, and as a result, a 1021-nucleotide covalently closed RNA molecule is produced (Figure 1A).

CircAFF1 is primarily described as a circRNA induced by hypoxia in HUV-EC-C and HBEC-5i cells. In these cells, circAFF1 inhibits cell proliferation, migration and tube formation of the endothelial cells by regulating YAP1 through a circAFF1/miR-516b/SAV1/YAP1 axis [34].

Analysis of RNA sequencing data previously produced in our laboratory [35] identified the presence of both circular and linear AFF1 isoforms in human primary myoblasts, which are considered our non-transformed control, as well as in RD and RH4 cell lines, representative of embryonal rhabdomyosarcoma (ERMS) and alveolar rhabdomyosarcoma (ARMS), respectively. To validate the presence of the back-splicing junction, we amplified it by RT-PCR with divergent oligos facing the BSJ and we submitted the amplicon to Sanger sequencing, which confirmed the presence of the back-spliced junction sequence (Figure 1B). To further confirm the circular nature of the detected RNA species, we amplified circAFF1 by qRT-PCR on RNA from RMS cells either untreated (RNaseR -) or treated with the RNase R exonuclease (RNaseR +), a 3′ exoribonuclease unable to degrade circular RNA [36]. As shown in Figure 1C, the addition of RNaseR affects neither the levels of the circAFF1 RNA nor the levels of other two well-known circular RNAs expressed in rhabdomyosarcoma, circHIPK3 and circVAMP3 [28], but it degrades the linear mRNA counterparts of the aforementioned RNAs.

To confirm the presence of this circRNA in rhabdomyosarcoma samples, we performed quantitative RT-PCR experiments on RNA obtained from primary RMS biopsies from patients and healthy tissues [37]; as can be observed in Figure 1D, ARMS samples tend to have increased amounts of circAFF1 compared to healthy tissues. Then, we tested the results in our representative cell line for each pathological and control condition (myoblasts, RD-ERMS and RH4-ARMS cells) by RT-qPCR. We observed a strong upregulation of circAFF1 in RD and RH4 metastatic cell lines compared to myoblasts with fold changes of 6.97 and 8.39, respectively, indicating that the upregulation of circAFF1 is maintained during the oncogenic process. On the other hand, the mRNA levels did not significantly change in RD cells or produced a much weaker increase in RH4 compared to the circular transcript (fold change = 1.45) (Figure 1E). These results were confirmed by the recent high-throughput data of circular RNAs in rhabdomyosarcoma, where circAFF1 is upregulated in RMS cells and is described as a circRNA not subjected to regulation through YTHDC1 m6A reader nor DDX5 helicase [35].

CircAFF1 is localised in the cytoplasm, as we demonstrated by cytoplasmatic/nuclear fractionation followed by qRT-PCR assay in RH4 cells (Figure 1F). Moreover, its sequence harbours a predicted ORF spanning its back-splicing junction; thus, we performed a sucrose gradient fractionation of cytoplasmic RD cell lysate to check its potential association with polyribosomes. However, we did not observe any enrichment in the polysome fractions (heavy and medium fractions; Appendix A); on the other hand, we found GAPDH mRNA to be associated with the heavy fractions of polyribosomes, suggesting that circAFF1 ORF is not translated, at least in our system.

### 3.2. CircAFF1 Knockdown Affects Adhesion-Related Pathways in ARMS Cells

ARMS is the most aggressive subtype of RMS, so we decided to first focus on the role of circAFF1 in the alveolar rhabdomyosarcoma cell lines using RH4 and RH30 cells, the most representative cell lines for this tumour, in which circAFF1 is abundantly expressed.

We knocked down circAFF1 in RH4 and RH30 cells with a siRNA targeting its BSJ (Figure 2A), and we used si-SCR as a control. Upon knockdown, we observed a strong downregulation (86.6% decrease) of the circular isoform with a small but significant downregulation (28.4% decrease) of the linear isoform in RH4 cells and a significant decrease (75%) in the circular isoform in RH30 cells, confirming the specificity of the selective downregulation (Figure 2A).

Then, we performed poly-A+ sequencing in both RH4 samples (si-SCR, si-circAFF1) to identify transcripts with a significant differential expression between the two conditions. Upon circAFF1 depletion, we found 530 deregulated genes (FDR < 0.05 and log2 fold change |0.58|) genes, of which 334 were downregulated and 196 were upregulated (Figure 2B, Appendix A).

We subjected the downregulated and upregulated subsets of genes to Gene Ontology (GO) term over-representation analysis. GO Biological Process term enrichment analysis on upregulated genes did not reveal any enriched category. However, downregulated genes showed enrichment (FDR < 0.05) in several processes related to heterotopic cell–cell adhesion, transport, extracellular processes, second messenger signalling and angiogenesis (Figure 2C). These data suggest a possible role of circAFF1 in the regulation of RH4 cell adhesion and matrix organisation as well as cell homeostasis by controlling transporters and signalling cascades.

We validated the RNA-seq results in both representative ARMS cell lines by selecting different deregulated genes involved in the abovementioned processes (Figure 2D). To ensure that the deregulation of these genes was specifically caused by circAFF1 knockdown, we added two different controls to the analysis. First, as we observed a decrease in AFF1 mRNA levels upon si-circAFF1 treatment, we designed a siRNA specifically targeting the linear isoform to exclude possible indirect/off-targeting effects caused by the absence of the linear isoform (Figure 2D, green columns). RH4 cells treated with si-AFF1 mRNA showed a downregulation of around 75% in the linear isoform and a reduction of 14% in the circRNA (Appendix A). As we can observe in Figure 2D, downregulation of the levels of the mRNA of the selected candidates was confirmed in both RH4 and RH30 only upon specific depletion of circAFF1; interestingly, we not only observed no downregulation of the selected candidates upon AFF1 mRNA depletion but in some cases, an upregulation, such as in the case of S100A2, RUNX2 and LRG5. CKAP5 was used as a random control for the unaltered genes from the RNA sequencing. As a second control to ensure the specificity of the downregulation of the aforementioned genes, we designed a second siRNA against the back-splicing junction sequence of circAFF1 (Appendix A); as shown in Appendix A, this second siRNA also produces a specific, albeit less efficient (fold change = 0.54), downregulation of the circular form of AFF1, which leads to a downregulation of SLC7A11, NOS1, RUNX2, TLR4 S100A2, LRG5 and GNB1 genes. Finally, we analysed the protein levels of some of these affected genes using protein extracts from RH4 cells treated with the si-SCR or upon specific depletion of circAFF1, and we could confirm that the downregulation at a protein level of NOS1, RNX2 (0.71 downregulation *p* value = 0.04 and 0.59 decrease *p* value = 0.0018, respectively) and the subtle but significant TLR4 downregulation (0.8 decrease, *p* value = 0.0058) emphasise the importance of circAFF1 in the regulation of the proteins involved in the heterotypic cell–cell adhesion (Figure 2E).

### 3.3. CircAFF1 Inhibits RH4 Cells Migration

As described by Wang and colleagues [34], circAFF1 inhibits the proliferation of endothelial cells by increasing the levels of the activator pathway of YAP1. However, in our system, circAFF1 depletion did not cause any alteration in either cell cycle, checked by FACS analysis (Figure 3A), or in the growing capacity of the cells, checked by growth curve assay (Figure 3B). Nevertheless, transwell migration assay showed that the knockdown of circAFF1, but not of AFF1 mRNA, led to 3-fold and 1.6-fold increases in the migration rate of RH4 and RH30 cells, respectively (Figure 3C), suggesting that circAFF1 controls the migration program in ARMS cells. The importance of the circAFF1 effect on migration was double-checked by measuring the RH4 migration rate using the second siRNA (Appendix A). As observed in Appendix A, si-circAFF1#2 also produces an increase in cell motility, confirming the specific involvement of the circular RNA in the described phenotype. To confirm these data, we performed wound healing assays in RH4 and RH30 cell lines upon treatment with si-SCR, si-circAFF1 and si-AFF1 mRNA. As observed in Figure 3D, only the specific downregulation of circAFF1 produces a significative decrease in the wound area 24 h after the scratch, confirming the importance of circAFF1 in the control of the migration program.

However, the mechanism by which Wang and colleagues proposed that circAFF1 controls this process is not altered in our system; we did not detect significant changes in either SAV1 or YAP1 in our RNA sequencing data, and miR-516b is not expressed in RMS [38]. Additionally, Western blot analysis performed on lysates from control, circAFF1- and AFF1 mRNA-depleted RH4 cells showed no significant changes in the protein levels of YAP1 after circAFF1 knockdown (Appendix A), suggesting an alternative mechanism of action for this circRNA.

Some of the genes downregulated upon circAFF1 knockdown are essential for migration and heterotypic cell adhesion. Hence, the downregulation of some of those genes might suffice to explain the observed phenotype in RH4 cells. S100A2 protein, a calcium-binding EF-hand motif protein, and TLR4, a member of the Toll-like receptor family, are two proteins that have been previously described as regulators of migration in diverse cancer cell lines [39,40].

As can be observed in Figure 3E,F, when performing a specific knockdown of TLR4 or S100A2, we noticed an increase in the RH4 cell migration rate, suggesting that the observed changes in cells motility upon circAFF1 knockdown are mediated through the downregulation of the genes involved in heterotypic cell adhesion such as TLR4 and S100A2.

## 4. Discussion

Rhabdomyosarcoma is one of the most widespread paediatric sarcomas, in which defects of the myogenic differentiation programme drive the pathological alteration of mesenchymal cells committed to skeletal muscle cell lineage [24].

The research of therapeutic targets and unique biomarkers is essential to improve the diagnosis and the outcome of the disease. CircRNAs are essential regulators of a variety of processes such as cell growth, metabolic reprogramming, angiogenesis and tumour onset and metastasis [18,19,20], and their description and categorisation are essential in research nowadays.

Here, we characterised circAFF1, a circular RNA abundantly expressed in alveolar rhabdomyosarcoma and embryonal rhabdomyosarcoma. The research on the effects of this particular circRNA encompassed until now the estimation of post-mortem interval [41], endothelial cell dysfunction [35], neuronal ferroptosis [42] and lens-derived pathologies [43], but never oncogenic processes. We demonstrated how the knockdown of circAFF1 can alter the migration capabilities of RH4 and RH30 cells, representative cell lines for alveolar rhabdomyosarcoma (ARMS), through the regulation of the cell adhesion programme genes, underlining the importance of this circRNA as a partial suppressor of a hallmark of cancer, such as the activation of cell migration and metastasis [44].

It is important to underline that the observed effects of circAFF1 are not produced by its linear counterpart, since the knockdown of AFF1 mRNA does not seem to replicate any of the observed effects. Previous reports on endothelial cells showed a potential link between this circular RNA and YAP/TAZ regulation using a microRNA sponge-mediated mechanism [34]. However, in rhabdomyosarcoma models, the observed phenotype is not produced through the Hippo pathway, nor the alteration of the cell cycle, but it is due to the downregulation of the mRNAs involved in heterotypic adhesion such as TLR4 and S100A2, able by themselves to mimic the phenotype when knocked down.

Although circAFF1 and its linear counterpart are both expressed in myoblasts, RD and RH4 cells, only circAFF1 is strongly upregulated in cancer cell types, which underlines its potential role in the transformation process. As shown in this manuscript, the overexpression occurs already in the primary tumour and is maintained in the metastatic representative cell lines of ARMS, suggesting its robust role during all oncologic development. The apparent strong involvement of circAFF1 as a key suppressor of some part of the migration capabilities is underlined by its early expression in primary tumours. Later in oncogenic development, cells acquire motility and migrate using other pathways that control motility. However, as demonstrated by the high levels of circAFF1 in metastatic cell lines RH4 and RH30, circAFF1 is still overexpressed and maintains its share of gene suppression genes on-check, indicating that RH4 and RH30 gained motility using one of the other pathways controlling the migration genetic program. It is noticeable that the alteration in the migration capabilities of rhabdomyosarcoma cells upon the depletion of circAFF1 is not accompanied by a change in the proliferation capabilities. Despite the crosstalk existing between cell motility and cell growth pathways, both can be altered independently (as observed, for example, in [45]) and both are considered by themselves independent hallmarks of cancer progression [44]. Exclusive alteration of the migration capabilities upon circAFF1 depletion, but not cell proliferation, clearly narrows the molecular pathways affected by this circular RNA and is in accordance with the transcriptomic data obtained after circAFF1 depletion.

Much more research is needed to unveil the molecular mechanism driving these effects; cirAFF1 might be acting as a ceRNA and sponging one or several microRNAs able to control the genes regulating cell migration at the same time or regulating a master regulator of motility processes. Another possibility might entail the sequestering of proteins necessary for the stability and regulation of the aforementioned genes. However, it must be remembered that these mechanisms of action have strong stoichiometric requirements and cannot be applied to a major fraction of circRNAs [46]. Another possible mechanism driving this regulation that circumvents stoichiometry issues might be circRNA–mRNA interactions, which recently have been demonstrated as a powerful and specific mechanism to regulate gene expression [16]. Moreover, a much wider analysis of the expression of circAFF1 is necessary to understand to which degree it is responsible for cell migration in several cancer stages of RMS and in other types.

It is also important to highlight that AFF1 locus rearrangements are responsible for many fusion-derived lymphomas in humans. It would be easy to suppose that in those cells where AFF1 expression is altered, the circAFF1 levels will also be changed, driving an alteration in the adhesion capabilities of the cells. Understanding the specific role of circAFF1 in the tumorigenic context will provide powerful tools for the diagnosis and prognosis of diverse cancer cell types, while the discovery of the molecular mechanisms controlled by circAFF1 may unveil new targets to tackle cancer progression.

## 5. Conclusions

In this work, we describe for the first time the involvement of circAFF1 RNA in a cancer model and how this circular RNA can regulate key factors controlling heterotypic cell adhesion. We show how alterations of circAFF1 RNA and downstream effectors impact the migration capabilities of cancer cells, emphasising the importance of this circular RNA in the tumorigenic process.

## Figures and Tables

**Figure 1 biomedicines-11-01893-f001:**
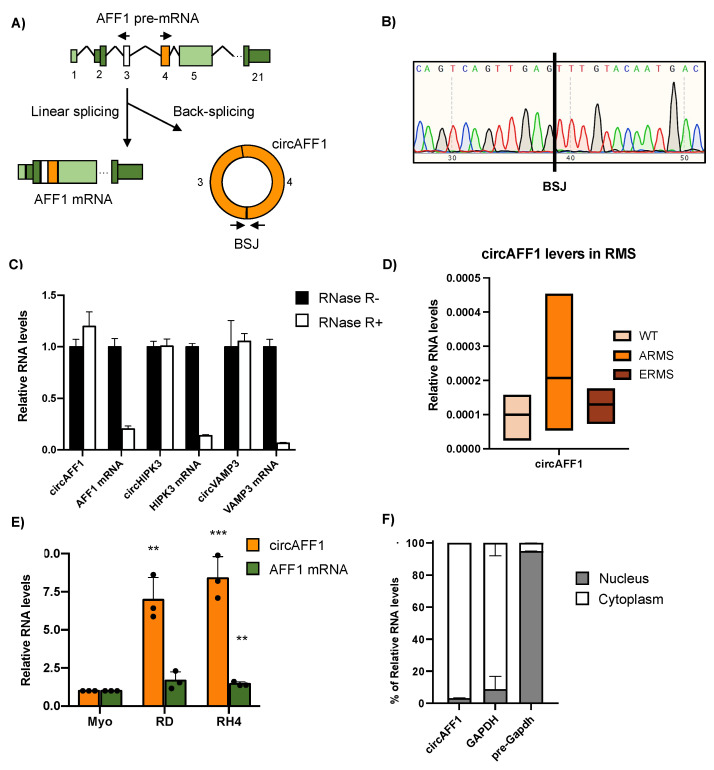
(**A**) Picture representing the splicing of AFF1 pre-mRNA when it undergoes canonical splicing to produce AFF1 mRNA or back-splicing to generate circAFF1. Divergent oligonucleotides to detect the circRNA are depicted as arrows. Back-splicing junction (BSJ) is indicated in the circAFF1 picture. (**B**) Sanger sequencing results from the circAFF1 RT-PCR amplification, depicting the BSJ. (**C**) Expression levels detected by RT-qPCR amplification of the selected circRNAs in control conditions (black) and after RNAse R treatment (white) of one biological replicate; data are represented as the average of fold changes ± standard deviation (SD) of three technical replicates. (**D**) Relative RNA levels normalised against GAPDH transcript of circAFF1 of at least three rhabdomyosarcoma (ARMS and ERMS) and health muscle samples (WT). (**E**) Relative RNA levels normalised against GAPDH transcript of circAFF1 (orange) and AFF1 mRNA (green) in myoblasts (Myo), RH4 and RD cells, discovered by qRT-PCR. Data are depicted as the average of fold changes ± SD of 3 experiments. Individual datapoints depicted as black dots. (**F**) Fraction of RNA allocation identified by qRT-PCR in subcellular compartments of RH4 cells for circular AFF1 RNA, GAPDH mRNA (characteristically cytoplasmatic) and GAPDH nascent RNA (typically nuclear). Results are depicted as the average distribution percentage ± SD of two biological replicates. Statistical analysis was calculated by the ratio vs. its experimental and tested by a two-tailed unpaired Student’s *t*-test. *p*-values were depicted as **: <0.01, ***: <0.001.

**Figure 2 biomedicines-11-01893-f002:**
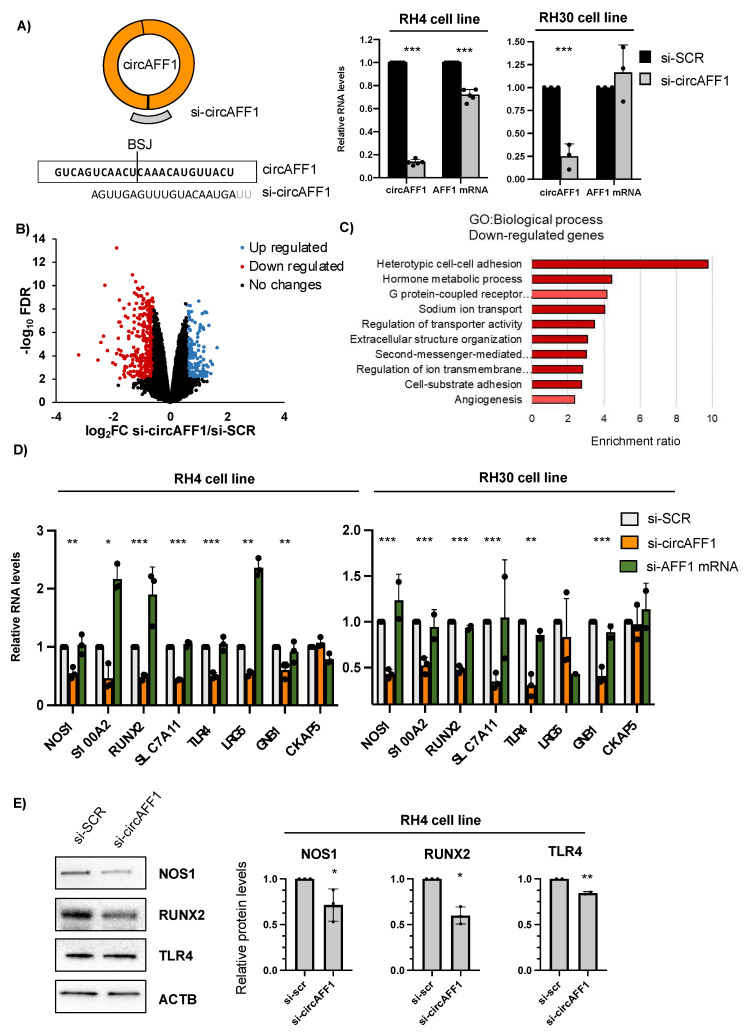
(**A**) Left panel: CircAFF1 knockdown strategy and the targeting sequence. Right panel: Relative levels of RNA normalised against GAPDH transcript detected by qRT-PCR of circAFF1 and AFF1 mRNA in RH4 and RH30 cells treated with si-SCR or si-circAFF1. Data are represented as the mean of fold changes ± standard deviation of at least 3 biological replicates. (**B**) Volcano plot showing genes differentially expressed in RH4 cells treated with si-circAFF1 vs. si-SCR. In blue are genes with a log2 fold change (FC) greater than 0.58 and an FDR lower than 0.05. In red are genes with a log2FC lower than −0.58 and an FDR lower than 0.05. (**C**) Gene Ontology (GO) term over-representation analysis of the downregulated genes upon depletion of circAFF1 in RH4 cells; dark red denotes FDR < 0.05 and light red denotes FDR < 0.1. (**D**) Relative levels of RNA normalised against GAPDH transcript detected by qRT-PCR of NOS1, S100A2, RUNX2, SLC7A11, TLR4, LRG5, GNB1 and CKAP5 upon si-SCR, si-circAFF1 (orange) or si-AFF1 mRNA (green) treatment in RH4 and RH30 cells. (**E**) Left panel: Protein levels of NOS1, RUNX2 and TLR4 in RH4 cells after si-SCR or si-circAFF1 treatment. Right panel: Relative to ACTB quantification of three independent experiments. Data are represented as the mean of fold changes ± standard deviation of biological replicates. Individual datapoints represented as black dots. Where statistical analysis was performed, the ratio of each sample vs. its experimental control was tested by a two-tailed unpaired Student’s *t*-test. *: *p*-value < 0.05, **: *p*-value < 0.01, ***: *p*-value < 0.001.

**Figure 3 biomedicines-11-01893-f003:**
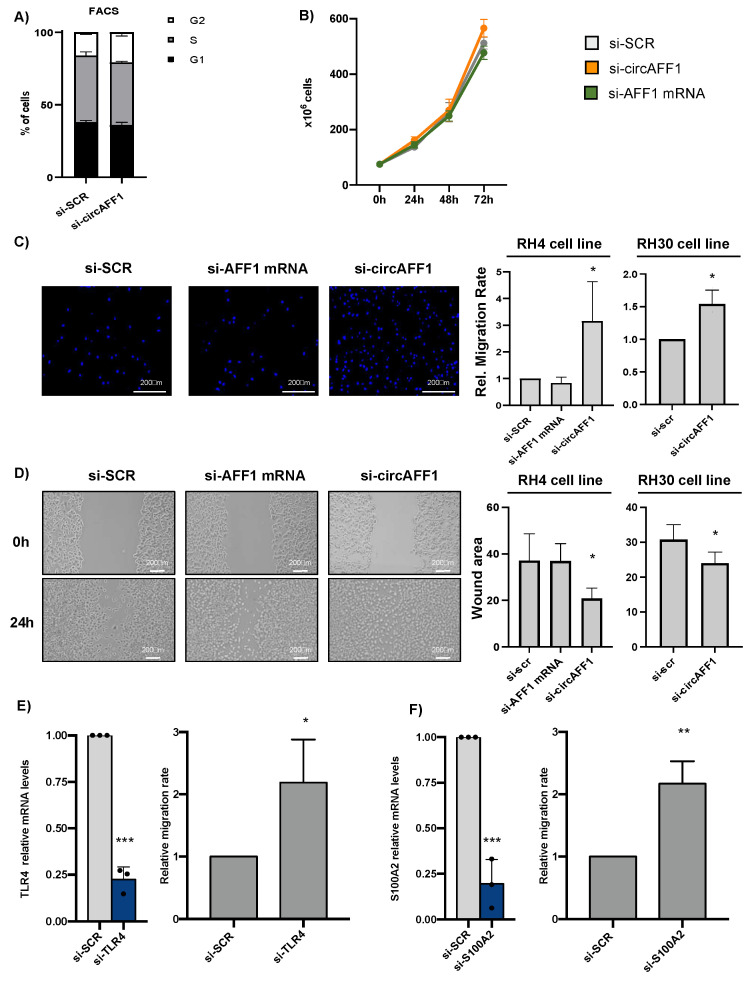
(**A**) FACS (flow cytometry) analysis to detect cell cycle alterations of RH4 cells after control treatment (si-SCR) or circAFF1 knockdown (si-circAFF1). Data are depicted as the average % cells in each cell cycle phase ± SD of two biological replicates. (**B**) Growth curve representing the number of cells in different time points upon si-SCR, si-circAFF1 and si-AFF1 mRNA treatment (depicted as the average number of cells and standard deviation of 3 biological replicates). (**C**) Right panel: Relative migration rate of RH4 and RH30 cells upon si-SCR, si-circAFF1 or si-AFF1 mRNA treatment. Data are depicted as the average of fold changes ± SD of 4 biological replicates. Left panel: Representative images of the migration assay coloured with DAPI. (**D**) Right panel: Non-covered area 24 h after monolayer scratch of RH4 and RH30 cells upon si-SCR, si-circAFF1 or si-AFF1 mRNA treatment. Data are presented as the mean of the difference between areas between d0 and d1 ± standard deviation of at least 3 biological replicates. Left panel: Representative images of the wound healing assay. (**E**) Left panel: Relative levels of RNA normalised against GAPDH transcript identified by qRT-PCR of TLR4 mRNA in RH4 lysates upon si RNA SCR (grey columns) or si-TLR4 (blue columns) treatment. Results are depicted as the average of relative increase ± SD of 3 experiments. Right panel: Relative migration rate of RH4 cells upon si-SCR or si-TLR4 mRNA treatment. Results are depicted as the average of relative changes ± SD of 3 independent experiments. (**F**) Same as D for S100A2 mRNA. Individual datapoints re represented as black points. Statistical analysis; the fold change of any given sample vs. control was assessed by a two-tailed unpaired Student’s *t*-test. *p*-values are depicted as *: <0.05, **: <0.01, ***: <0.001.

## Data Availability

RNA sequencing raw data have been deposited in Gene Expression Omnibus (GEO) with the accession code GSE214354.

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
