# Peer review of "CircAFF1 Is a Circular RNA with a Role in Alveolar Rhabdomyosarcoma Cell Migration"

_biomedicines, 2023, doi:10.3390/biomedicines11071893_

Round 1

Reviewer 1 Report (Previous Reviewer 2)

The authors have responded to my previous comments in two rounds of revisions and the manuscript has been significantly improved after resubmission.

Minor editing of English language may be required.

Author Response

We would like to thank the reviewer for the positive review and the help and input received during the reviewing process.

Reviewer 2 Report (Previous Reviewer 1)

The authors did not/don't want to study the underlying mechanisms of CircAFF1 in Alveolar rhabdomyosarcoma. The study is incomplete unless the the relationship between the CircAFF1, RUNX2 and NOS1 is experimentally explained. The discussion is biased and illogical, conlusion is baselss. I am sorry, I vote against the study this time and ask no more revisions.    

Author Response

We strongly disagree with the opinion of the reviewer for the reasons mentioned in the earlier parts of the revision process.

Reviewer 3 Report (New Reviewer)

This is an interesting manuscript delineating the role of circular RNA CircAFF1 in rhabdomyosarcoma. The authors demonstrated over expression of this circular RNA in few RMS cell lines. The overexpression was associated with decrease in cell mobility, cell migration, and possibly genes associated with cell adhesion and transport. The manuscript is well-written, illustrated and clear. However, there are few areas that are ambiguous and need improvement clarification:

1.  The abstract is short and needs more information, at least stating that experiments were performed on cell lines.

2. Numerous circular RNAs are overexpressed in rhabdomyosarcoma, as illustrated in the new paper by Datillo et al (Nat Commun 2023; 14:1898). I am curious about the reason why the authors selected circAFF1 for their studies. Was this particular circRNA part of the groups mentioned by Datillo's article? Was there any previous finding about its role in RMS? 

3. Lines 180-182 and Fig 1D state that RT-PCR experiments were performed on patients samples. I do not find description of this patient population and tumors involved. This should have been listed in a table with separate heading in the methods section.

4. The studied cell lines are few and are all derived from patients with metastasis. That might affect experiments and explain the dysregulation of genes associated with cell adhesion and transport. I recommend studying the expression of this circRNA in patients samples containing two types of tumors: localized tumors and tumors with metastasis.

5. The effect of circAFF1 on the general outcome of RMA is not clear. The overexpression leads to arrest in cell migration and mobility. How do the authors justify this overexpression in metastatic cell lines? Do the authors claim that this circRNA has tumor suppressive properties?  If so, this conclusion should be clearly mentioned in the conclusion section. 

Author Response

This is an interesting manuscript delineating the role of circular RNA CircAFF1 in rhabdomyosarcoma. The authors demonstrated over expression of this circular RNA in few RMS cell lines. The overexpression was associated with decrease in cell mobility, cell migration, and possibly genes associated with cell adhesion and transport. The manuscript is well-written, illustrated and clear. However, there are few areas that are ambiguous and need improvement clarification:

We would like to thank the reviewer for the positive review and for the indications to improve the manuscript. We modified the article to clarify some points following his/ her suggestions. Hereby a detailed response to the reviewer’s concerns.

  1. The abstract is short and needs more information, at least stating that experiments were performed on cell lines.

Following the reviewer's suggestions, we added information to the abstract to help to understand the content of the article (lines 14-17)

  1. Numerous circular RNAs are overexpressed in rhabdomyosarcoma, as illustrated in the new paper by Datillo et al (Nat Commun 2023; 14:1898). I am curious about the reason why the authors selected circAFF1 for their studies. Was this particular circRNA part of the groups mentioned by Datillo's article? Was there any previous finding about its role in RMS?

CircAFF1 appeared in the original RNAseq data comparing myoblast and RMS cell lines which also was seminal for Dattilo’s work. In parallel to the study of the effect of m6A readers performed by Dattilo, we selected a few of those circRNAs to study the putative phenotypic effects upon the depletion of those circular RNAs in RMS as none of them was described to have a role in RMS before. We selected circRNAs upregulated in RMS without a parallel upregulation of the linear mRNA counterpart (corresponding to the term discordant circRNAs in Datillo’s article). CircAFF1 was one of the circular RNAs that were among the most upregulated in rhabdomyosarcoma without having an increase of its linear mRNA counterpart.

We added a line in the article to underline the findings about the description and methylation status described in Dattilo et al this circular RNA is also found in Dattilo’s data (lines 197-200).

  1. Lines 180-182 and Fig 1D state that RT-PCR experiments were performed on patients samples. I do not find description of this patient population and tumors involved. This should have been listed in a table with separate heading in the methods section.

We apologise for this lack of information and the methods section has been updated as the reviewer requested (Lines 82-89).

  1. The studied cell lines are few and are all derived from patients with metastasis. That might affect experiments and explain the dysregulation of genes associated with cell adhesion and transport. I recommend studying the expression of this circRNA in patients samples containing two types of tumors: localized tumors and tumors with metastasis.

We share the concern with the reviewer about this point, and it would be very interesting to follow the expression levels and effects of this circRNA in different stages of the sickness. 

Unfortunately, the classic rhabdomyosarcoma cell lines are mostly derived from metastasis (RP Hinson et al. 2013, 10.3389/fonc.2013.00183), and we do not have immediate access to other cell lines. 

RD and RH4 might represent an advanced stage of the illness and might already have altered cell adhesion and transport capabilities. Nevertheless, the observation that in those cells the down-regulation of circAFF1 increases cell motility and provokes the downregulation of the involved genes provides proof of the molecular involvement of this circular RNA in migration, independently of the basal alteration present in RH4 and RH30 cell lines.

Our tumour samples come from primary tumours, with no metastasis involved and due both to the time constraints during revision and, above all, due to the scarceness of the requested materials, we can’t obtain samples from matched metastasis tumours. 

However, our results indicate not only that circAFF1 regulates a set of genes affecting the migration program, hence being a partial suppressor of the migration program, but that it is already expressed in primary biopsies, in comparison with healthy tissues. The increase of circAFF1 observed in primary tumours is about 2 folds with respect to the healthy controls. This increase is about 7 folds when looking at circAFF1 expression levels in the metastasis-derived cell lines.

Migration can be driven through several diverse pathways. CircAFF1 might be only controlling a small part of this cellular program, so it is plausible that it did not participate in the metastatic process of the primary tumours from whose metastasis we derived RH4 and RH30. Indeed, the migration of these cell lines can be further altered by circAFF1 downregulation, indicating that in those cells circAFF1 is still exploiting its function.

In line with this hypothesis, the upregulation of circAFF1 in metastasis-derived cell lines could be read as acompensatory mechanism to keep its part of the migration program on-check. 

We had no intention to indicate cirAFF1 as directly responsible for the general metastatic process in RH4 and RH30 cell lines, but to point out the fact that it can be part of a motility pathway that is possibly still untouched in our cells.

We apologise for the lack of clarity on this point and we added some lines in the text to help clarify this topic (lines 188 & 194, 358-361, 387).

  1. The effect of circAFF1 on the general outcome of RMA is not clear. The overexpression leads to arrest in cell migration and mobility. How do the authors justify this overexpression in metastatic cell lines? Do the authors claim that this circRNA has tumor suppressive properties?  If so, this conclusion should be clearly mentioned in the conclusion section. 

We apologize for the lack of clarity on this point; as explained in point 4 we suggest circAFF1 as a gene controlling the regulation of a set of genes that are involved in the migration program, hence being a partial suppressor of the migration program. Migration can be driven through several diverse pathways. During tumour development cells metastasize using one of the multiple pathways that lead increase of motility, but our circRNA is even higher expressed, possibly as a compensatory mechanism to keep its share of the migration program on-check. Prove of that is the fact that downregulation of circAFF1 promotes higher motility even in RH4 and RH30 cell lines, indicating that those cells gained motility overriding one of the other pathways that lead to metastasis.

We added some lines to the discussion clarifying this point (lines 18, 347, 362-368)

This manuscript is a resubmission of an earlier submission. The following is a list of the peer review reports and author responses from that submission.

Round 1

Reviewer 1 Report

In this study, authors analyzed the role of a circRNA CircAFF1 in the proliferation of alveolar rhabdomyosarcoma cells. First, they found it overexpressed in the alveolar rhabdomyosarcoma cells and knockdown of this CircAFF1 impact the migration of the cells. Although not many studies have reported the role of CircAFF1 in cancer, especially in alveolar rhabdomyosarcoma but authors still did not fully explore its functions. They knocked it down using siRNA and found that the proliferation and migration of the cells were increased. Additionally, they performed RNA sequencing and detected differential expression of some genes. A lot needs to be done, such as phenotypic experiments like an invasion, wound healing, cell cycle, cell death, xenograft, etc. Regarding pathways analysis or target genes, they need to pulldown CircAFF1, do MS, and perform further mechanistic studies. Major question how CircAFF1 affects the adhesion-related pathways or others? Additionally, they should study the regulation of CircAFF1 and its targets. 

Reviewer 2 Report

In the article, the researchers describe the role of circAFF1 in alveolar rhabdomyosarcoma cell migration. The study is interesting and suggests direction for further studies. Nevertheless, there are major concerns about the studies that need to be improved before publication.

The font on the figures is illegible, so it is difficult to interpret and understand the presented results.

The discussion needs to be significantly improved in the context of the current literature. The discussion section is the week point of the article.

Silencing of circAFF1 exerts effects only on migration of the cells and does not affect proliferation. Therefore circAFF1 is not a crucial factor in rhabdomyosarcoma progression. More experiments investigating that phenomenon need to be performed in vitro. It would be also interesting to add the results from the studies in vivo.

The experiments with circAFF1 silencing were performed using only one alveolar rhabdomyosarcoma cell line – RH4. To draw conclusions about circAFF1 role in alveolar rhabdomyosarcoma, the experiments need to be repeated using at least one more alveolar rhabdomyosarcoma cell line.

The authors should suggest and investigate the mechanism of action of circAFF1 in rhabdomyosarcoma.

There also a doubt in selection of constitutive gene for analysis of qPCR data. Please explain why it was selected.

The authors should add the statement about analysis of STR profile and Mycoplasma testing in the cell lines.

The most important results from RNA sequencing should be also analyzed at protein level using Western blot.

Round 2

Reviewer 1 Report

I still think the study is incomplete; no evidence is provided for the mechanisms and pathways. I suggest authors should perform detailed research and resubmit a new article. The logic is unclear, some results are poorly explained, and experiments are poorly designed. Authors should focus on how CircAFF1 regulates NOS1 and RUNX2 and further explain the relationship between the phenotype. At least they should perform Co-IP/RNA-IP for the relationship between NOS1 and RUNX2. If there is no direct relationship, then find intermediates that interact with CircAFF1 and regulate it/regulated by it or regulate NOS1 and RUNX2. Further, alter the expression of these factors, etc., and also perform rescue experiments. 

Reviewer 2 Report

The article is interesting and the authors have significantly improved the manuscript by their corrections. Nevertheless, before final decision, there are several issues that should be improved.

 The authors should write about RH30 cell lines in the introduction section together with description of RH41 and RD cell lines.

 RH30 cell line should be added to the subsections in the materials and methods (each technique).

·       Molecular mechanisms should be described more deeply in the discussion section with focus on the regulated genes presented in the results sections.

·       TLR4 seems to not be regulated according the Western blot image - please be more careful about your statement in the manuscript text.

·       Please discuss the lack of the effect on  proliferation in the manuscript text (discussion section).

·       Please correct the typo errors in the manuscript.